# Two novel genomes of fireflies with different degrees of sexual dimorphism reveal insights into sex-biased gene expression and dosage compensation

Ana Catalán [1] ✉, Daniel Gygax[1,2], Leticia Rodríguez-Montes [3], Tjorven Hinzke[4,5], Katharina J. Hoff [6] & Pablo Duchen [7]

Sexual dimorphism arises because of divergent fitness optima between the sexes. Phenotypic divergence between sexes can range from mild to extreme. Fireflies, bioluminescent beetles, present various degrees of sexual dimorphism, with species showing very mild sexual dimorphism to species presenting female-specific neoteny, posing a unique framework to investigate the evolution of sexually dimorphic traits across species. In this work, we present novel assembled genomes of two firefly species, *Lamprohiza splendidula* and *Luciola italica*, species with different degrees of sexual dimorphism. We uncover high synteny conservation of the X-chromosome across ~ 180 Mya and find full X-chromosome dosage compensation in our two fireflies, hinting at common mechanism upregulating the single male X-chromosome. Different degrees of sex-biased expressed genes were found across two body parts showing different proportions of expression conservation between species. Interestingly, we do not find X-chromosome enrichment of sex-biased genes, but retrieve autosomal enrichment of sex-biased genes. We further uncover higher nucleotide diversity in the intronic regions of sex-biased genes, hinting at a maintenance of heterozygosity through sexual selection. We identify different levels of sex-biased gene expression divergence including a set of genes showing conserved sex-biased gene expression between species. Divergent and conserved sex-biased genes are good candidates to test their role in the maintenance of sexually dimorphic traits.

Females and males tend to show mild or extreme differences in their biological and ecological requirements[1,2]. Sexually dimorphic traits usually arise to accommodate and optimize a sex's specific needs, sometimes through sex-specific genetic variants which usually lie on the sex chromosomes, but mostly through gene expression divergence between the sexes[3,4]. Differentially expressed genes between the sexes are good candidates for the resolution of sex-specific phenotypes, and thus the resolution of sexual antagonism[5]. Depending on the studied taxa, sex-biased genes can have high gene expression turnover across species, which might suggest strong sex- and species-specific selective pressures acting on gene expression[6,7]. Less

attention might have been given to conserved sex-biased expressed genes across species[8–10], even when this set of genes can provide us with knowledge about common inter-species processes maintaining sexual dimorphism. At the protein level, sex-biased genes can have rapid evolutionary rates[11,12] either as a response to high sexual selective pressures[13] or resulting from a relaxation of purifying selection[14].

Fireflies, fluorescent beetles of the family Lampyridae, show different degrees of sexual dimorphism, with some species presenting female neoteny and fully developed males, whereas others have only mild differences between the sexes[2,15]. For example, in the small European firefly *Lamprohiza*

[1]Ludwig-Maximilians-Universität Munich, Division of Evolutionary Biology, Großhaderner Straße 2, Planegg-Martinsried, Bavaria, 82152, Germany. [2]Helmholtz Center Munich, Helmholtz Pioneer Campus, Ingolstädter Landstraße 1, Munich, Oberschleißheim, 85764, Germany. [3]Center for Molecular Biology of Heidelberg University (ZMBH), DKFZ-ZMBH Alliance, D-69120 Heidelberg, Germany. [4]Institute of Microbiology, Department of Microbial Physiology and Molecular Biology, University of Greifswald, Greifswald, Germany. [5]Department of Pathogen Evolution, Helmholtz Institute for One Health, Greifswald, Germany. [6]University of Greifswald, Institute for Mathematics and Computer Science, Walther-Rathenau-Str. 47, 17489 Greifswald, Germany. [7]Institute of Organismic and Molecular Evolution, Johannes Gutenberg University of Mainz, Hanns-Dieter-Hüsch-Weg 15, 55128 Mainz, Germany. ✉e-mail: ana.catalan@gmail.com

*splendidula*, adult females are neotenic with reduced wings and heads, whereas males are fully developed adults able to fly (Fig. 1). On the other hand, the Italian firefly *Luciola italica* shows milder sexually dimorphic traits, where both sexes are fully winged, both show a flashing behavior during mating, and the head capsule of females is only mildly reduced when compared to males (Fig. 1). The level of sexual dimorphism across fireflies and other insects has been associated with the type of mating system and the presence of male spermatophores (nuptial gifts), posing a tradeoff between dispersion and oviposition[2,15]. *L. italica* and *L. splendidula* diverged ~ 140 mya[16], into different degrees of sexual dimorphism, and provide us with a comparative framework to study sex-biased transcriptome variation and conservation between species.

Sex chromosomes are highly differentiated in beetles, including fireflies. Most of the firefly species studied have an XO sex determination system, where males are the heterogametic sex[17,18]. The full degradation of one of the X homologs, causes gene expression imbalance between the X-chromosome and the autosomes and between the X-chromosome copy number between the sexes[3]. In species where X (or Z) chromosome dosage compensation has evolved, it has been hypothesized that highly deleterious effects are produced when the gene expression dosage of interacting genes between the autosomes and the X is shifted or disrupted[19,20]. Similarly, shifts in X-linked gene expression between female and males with phenotypic effects, might be under strong selection against variation, promoting gene expression balance at the chromosome[21] or at the gene level[22]. Overall, expression imbalance between the autosomes and between the sexes, is hypothesized to be highly detrimental, so much, that highly complex mechanisms have evolved to restore dosage balance in some taxa in an independent manner[23,24]. Some beetles show full X DC[25], but how conserved is DC within beetles and what is its status in fireflies still needs to be explored.

Here, we present two new chromosome scaffolded high quality genome assemblies of *L. splendidula* and *L. italica*, genomic resources that will contribute to the investigation of the evolution of sex-biased gene expression, and will additionally serve as genomic resources for micro- and macroevolutionary studies. Moreover, we generated transcriptome data from heads and abdomens of both sexes, in order to investigate the genes that are sexually dimorphic expressed and to investigate what is the level of gene expression conservation/divergence between these species. We also uncover full dosage compensation in these fireflies' species, opening the question of how conserved is DC in this group.

## Results

### Luciola italica and Lamprohiza splendidula genomes

We used a combination of long and short read technologies (Nanopore + Illumina) complemented with Arima Hi-C (Fig. S1) to generate chromosome level genome assemblies for *L. splendidula* and *L. italica*. The genomes of these species have different characteristics in size, heterozygosity levels, chromosome number, and percentage of repetitive elements (Table 1). The lower heterozygosity found in *L. splendidula* in comparison to *L. italica*, could be explained by its putative lower migration rates, since dispersion is expected to be reduced in *L. splendidula*, where neotenic females do not move in adulthood. *L. italica*, a species native to southern Europe, has recently expanded its distribution northern of the Alps (up to Lausanne and Basel, Switzerland)[26]. Since *L. italica*'s placement in a phylogenetic context was missing, we used anchored hybrid enrichment (AHE) loci[27] to place *L. italica* in the current firefly phylogeny. *L. italica* was placed as a sister species to *Abscondita cerata* and to *Lucila* sp. 2, a specimen which was collected in Vietnam (Fig. S2). This result highlights the need of a more complete sampling of firefly species, which will lead to a better understanding of their evolutionary history at a phylogenomic scale.

We identified the X-chromosome in both species by identifying contigs having an m:f coverage close to a ratio of 0.5 (Fig. S3), and corroborated these results with Hi-C structural linkage. *L. italica*'s X-chromosome is 60% larger than that of *L. splendidula*. The X-chromosomes comprised between 6.3% (*L. italica*) and 7.1% (*L. splendidula*) of their total genome size (Table 1). Additionally, we verified the identity of the X-chromosome by using homology to other elateroid beetles, showing high synteny conservation across ~ 180 MY (Fig. 2). We also identified autosomal

**Fig. 1 | Stereo microscope images of collected specimens.** Dorsal and ventral view of female and male individuals of *Luciola italica* (left) and *Lamprohiza splendidula* (right).

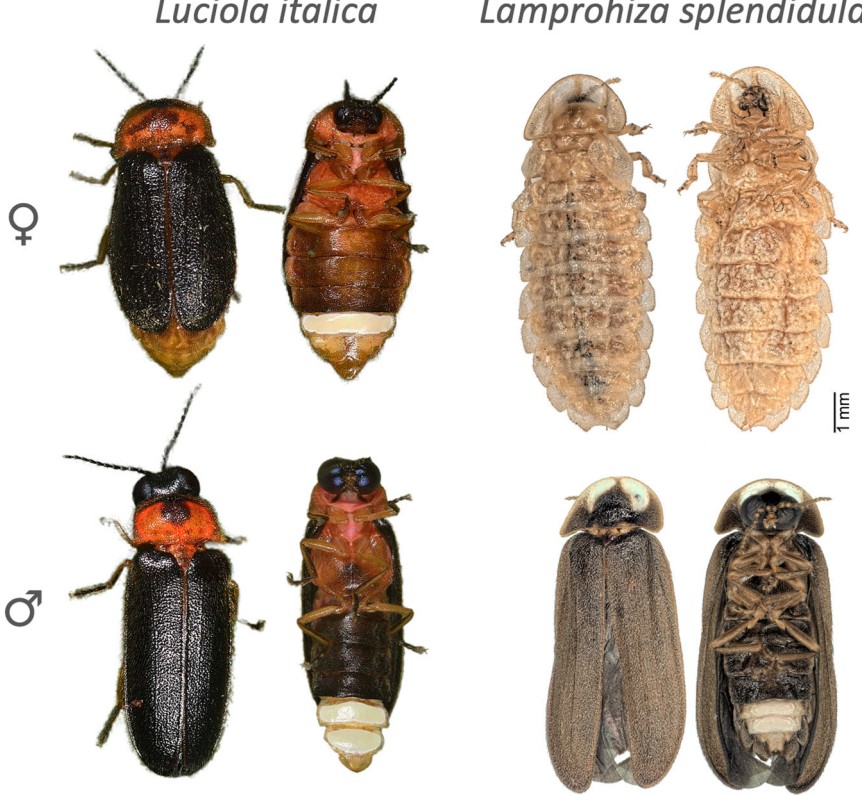

*Luciola italica*    *Lamprohiza splendidula*

**Table 1 | General genome and annotation statistics**

| Statistic | *Lamprohiza splendidula* | *Luciola italica* |
|---|---|---|
| Genome size (bp) | 637,778,614 | 1,210,014,512 |
| Repeat content | 52.18% | 58.29% |
| Genome heterozygosity | 0.27% | 0.66% |
| CG content | 34.10 | 33.38 |
| N50 (bp) | 77,400,372 | 126,850,438 |
| L50 (bp) | 4 | 7 |
| Number of scaffolded contigs | 8 + 1X | 9 + 1X |
| Total number of contigs | 350 | 1891 |
| BUSCO scores | [a]C:96.6% [S:95.4%, D:1.2%], F:2.1%, M:1.3% | [a]C:98.3% [S:94.7%, D:3.6%], F:1.2%, M:0.5% |
| Number of annotated genes | 33,764 | 38,309 |
| X-chromosome length (bp) | 45,498,505 (7.1%) | 75,711,303 (6.3%) |

[a]*C* complete BUSCO genes, *S* single copy genes, *D* duplicated genes, *F* fragmented genes, *M* missing genes. The dataset Insecta (n = 1367) was used in both species.

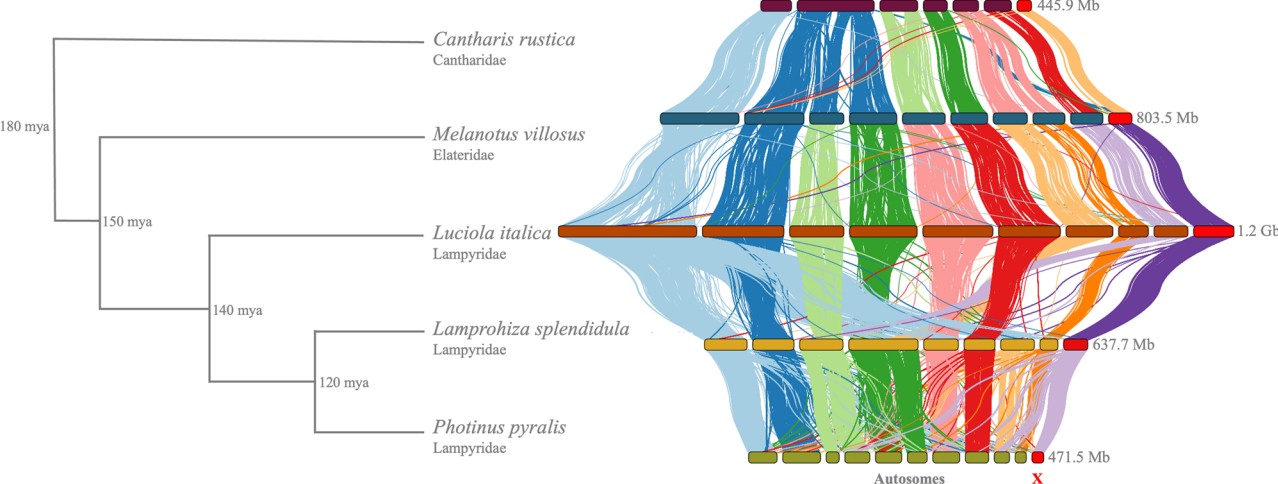

**Fig. 2 | Synteny comparison between five genomes of elateroid beetles.** Left: Scheme of divergence times and phylogenetic relationships of the species tested[16,53]. Right: Horizontal bars represent chromosomes and vertical connecting lines represent synteny anchors between genomes.

contributions (or losses) to the X-chromosome, involving different autosomes (Fig. 2, Figs. S4–S6). Within some insect groups, synteny across the X-chromosome has been reported to be highly conserved[25,28], whereas in other groups such as flies or butterflies, the evolution of sex chromosomes appears to be more dynamic[29,30]. At the autosomal level, we uncovered several autosomal fusion and fission events (Fig. 2). For example, in *Melanotus villosus* (Elateridae), chromosomes 2, 3 and 4 are fused into *Cantarhis rustica*'s (Cantharidae) chromosome 2. In *L. italica* chromosome 1 maps at least into four chromosomes in *L. splendidula*, whereas chromosome 4 and 6 in *L. italica* is fused to chromosome 4 of *L. splendidula*.

**Dosage compensation in fireflies**

Sex chromosome dosage compensation has evolved independently in many taxa as a mechanism to equalize gene expression in the heterogametic sex and it is unknown whether fireflies show DC. To investigate DC in *L. splendidula* and *L. italica*, we calculated the female to male (f:m) gene expression ratios for each chromosome, as well as the X to autosomal gene expression ratios (X: A) in females and males. We found full dosage compensation ($XX_f:X0_m = 1$) and balance $XX_f:AAf => 1$, $X0_m:AA_m => 1$ in both firefly species (Fig. 3).

When X: A expression ratios in *L. italica* were assessed, we observed a significant overcompensation of the X-chromosome in the heads and abdomens in both sexes (Wilcoxon-test, *p*-value < 0.0001) (Fig. 3, Fig. S7). This result suggests that a mechanism leading to upregulation of the male's

X chromosome might be involved in the equalization of chromosomal gene expression.

**Comparative sex-biased gene expression**

Species showing different degrees of sexual dimorphism provide us with a comparative framework to study the genetic basis of sexually dimorphic traits. Here, we use two species of fireflies with different degrees of sexual dimorphism: *L. italica* showing mild sexual dimorphism and *L. splendidula* with extreme sexual dimorphism, where the females are neotenic (Fig. 1). We evaluated the expression of sex-biased genes in the heads and abdomens of adult individuals of *L. italica* and *L. splendidula*. We had at least five replicates per sample type to ensure a confident assessment of sex-biased gene expression (Supplementary Data 1–4). Heads and abdomens of both firefly species showed different patterns of sexual dimorphism. In heads, the neotenic species *L. splendidula*, showed ~ 70% more sex-biased genes than *L. italica*, being male-biased genes in *L. splendidula* 54% more abundant than female-biased genes. A flipped pattern was observed in abdomens, where *L. italica* showed 34% more sex-biased genes than *L. splendidula*. Also, female-biased genes in *L. italica* were 1.4% more abundant than males-biased genes (Fig. 4A).

We then classified sex-biased gene expression in three expression categories: 1. Weakly sex-biased, 2. Intermediate sex-biased and 3. Sex-specific expressed (see methods). The largest category for both species and body parts were weakly sex-biased expression, followed by intermediately

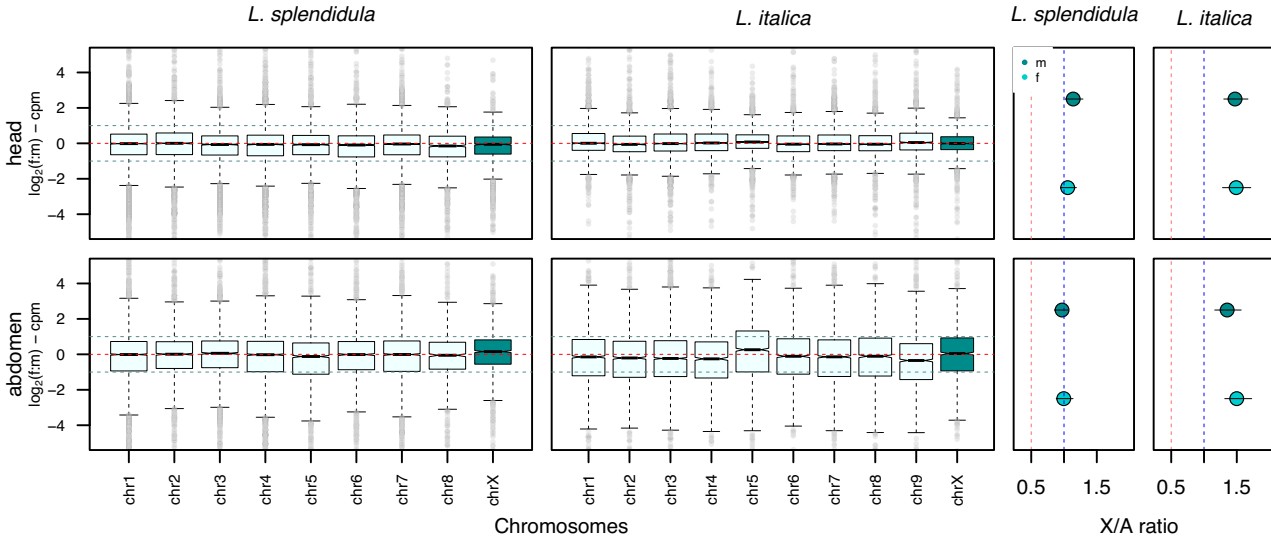

**Fig. 3 | Dosage compensation status in two firefly species in heads and abdomens.** Left: Boxplots show female to male (f:m) ratio distribution of expressed genes per chromosome. Chromosome X is marked with dark blue. Right: Dot plots showing X to autosomal (X: A) gene expression ratios in female and males. Bars in dot plots represent 95% confidence intervals drawn from 10,000 bootstraps.

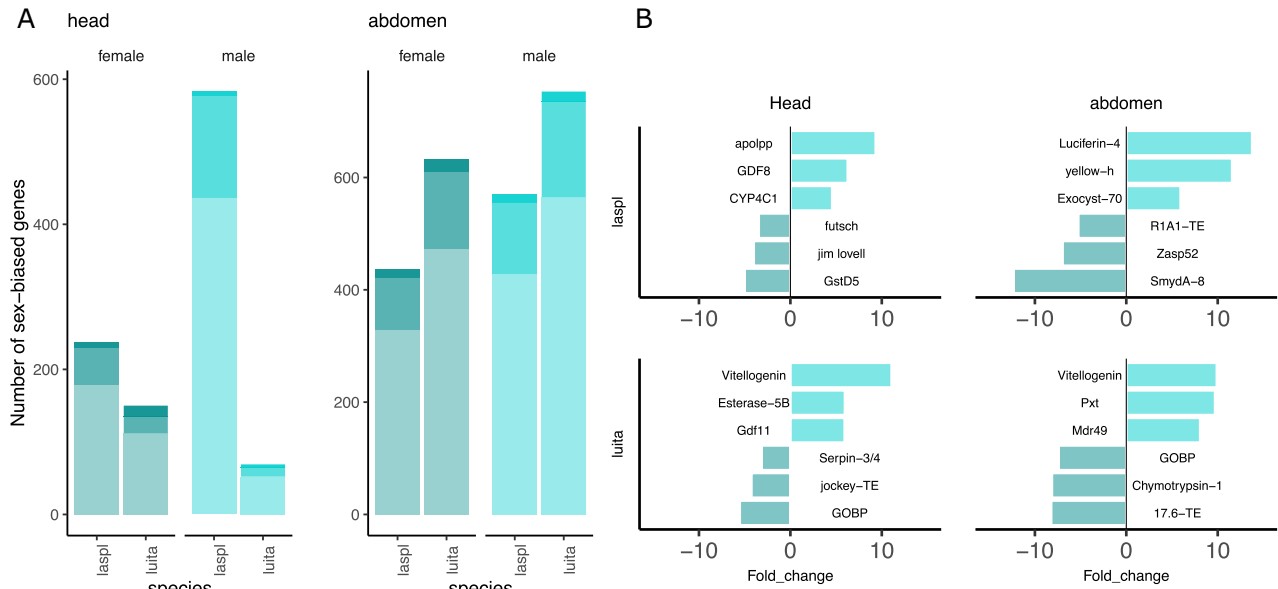

**Fig. 4 | Number of sex biased genes categorized by sex-biased fold difference in heads and abdomens. A** Cutoff for sex-biased gene classification was set up at FDR < 0.05 and logFC>2. Light blue: weakly sex-biased genes (logFC> first quantile & logFC<third quantile). Medium blue: intermediate sex-biased (logFC>third quantile & logFC <(third quantile+max/2). Deep blue: sex-specific (logFC>(third quantile max/2). **B** Top three sex-biased genes with an annotation match. Dark blue: male-biased, light blue: female-biased.

sex-biased and genes showing sex-specific expression was the smallest category (Fig. 4A). Sex-specific expressed genes are probably involved in sex-specific gene regulation. Sex-specific expressed genes were identified in abdomens and heads of both species, and in *L. italica*'s heads, these were significantly enriched on the X-chromosome (Fisher Exact Test: *p*-value 6.095e-08).

Some of the genes that show the strongest female-biased expression in *L. splendidula* heads are involved in size regulation (*apolpp*, GDF8)[31,32] and the metabolization of xenobiotic substances (Cytochrome P450 – CYP4C1)[33]. *L. splendidula* male heads also showed male-biased expression in genes involved in detoxification (GstD5)[34], as well as in genes involved in courtship behavior (*jim lovell*)[35]. In *L. italica*'s female heads, genes involved in reproduction (*vitallogenin*)[36] and detoxification (Esterase-5B)[37] were

found. In abdomens, *luciferin-4*[38] showed female-specific expression and the *yellow-h*[39] gene, involved in sex-specific maturation, showed strong female-biased expression in *L. splendidula*. On the other hand, genes involved in flight ability were identified to be strongly male-biased in *L. splendidula*. In *L. italica*, expected female-biased genes such as *Peroxinectin-like*[40] gene involved in oogenesis were found, while in males, genes involved in digestion (*Chymotrypsin-1*)[41] hint at an active digestion system in adults of this species (Fig. 4B).

Often, the state of sexually dimorphic expression of orthologous genes tends to show high turnover rates between species and is thought to be the result of species-specific or sex-specific selection[6,7]. *L. splendidula* and *L. italica* diverged ~ 140 Mya, which would allow enough evolutionary time for species-specific sex-biased gene expression. Conversely, genes showing

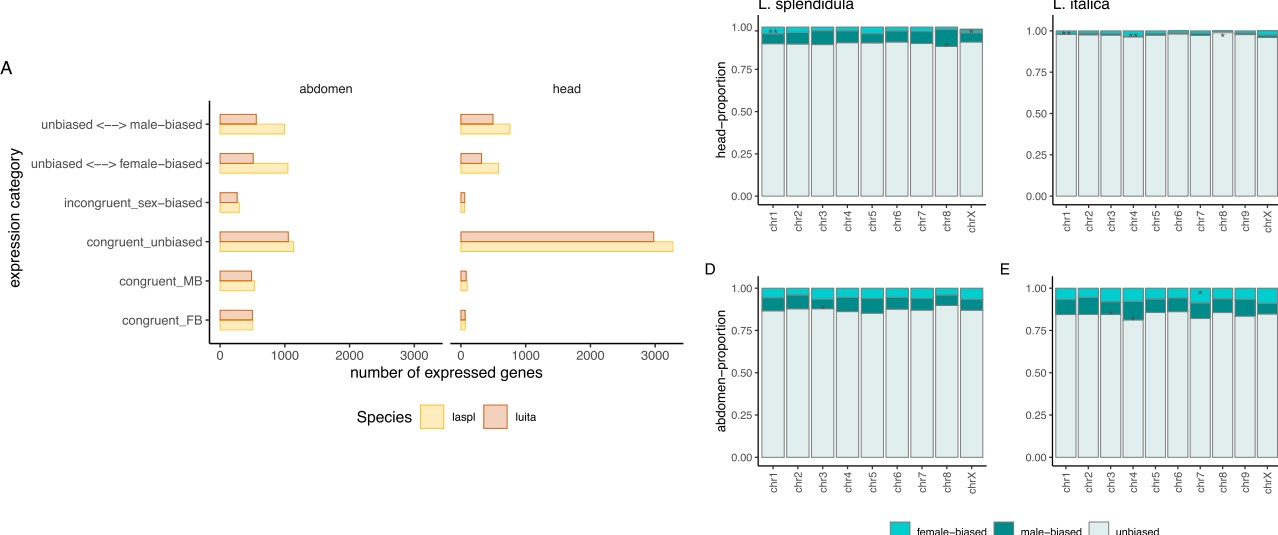

**Fig. 5 | Congruence levels of sex-biased expression and chromosome enrichment.**
**A** Congruent sex-biased genes refer to genes that are sex-biased in the same direction in both species. Incongruent sex-biased genes refer to genes that are sex-biased in the opposite direction. The two headed arrow depicts transitions from unbiased to sex-biased or vice versa. **B–E** Chromosomal enrichment analysis from sex-biased genes. Stars depict significant levels as tested with a Fisher's exact test: 0.0001 '***', 0.001 '**', 0.01 '*'.

concordant sex-biased expression, might be involved in conserved pathways maintaining sexually dimorphic traits across deeply diverged species. We looked at the degree of congruency/incongruency in sex-biased gene expression between the two species and calculated the number of genes with an agreeing/disagreeing expression patterns. The majority of sex-biased genes presented incongruent gene expression, where 17–24% of abdomen sex-biased expression was not congruent between species. In heads incongruent sex-biased expressed genes represented 8–11% of genes where an orthologous was found. In abdomens, between 4–7% of sex-biased expressed genes showed congruent sex-biased expression. Interestingly, the percentage of sex-biased congruent expression in heads decreased to 0.1–0.3% (Fig. 5A). We observed shifts from unbiased to sex-biased, where the proportion of changes from unbiased to female or male-biased is not significantly different between the sexes (one-proportion Z-test, not significant) (Fig. 5A), hinting that neither sex is driving an excess of sex-biased gene expression.

Sex-biased genes are often observed to be depleted or enriched on the X-chromosome[42,43]. A sex conflict resolution scenario has been proposed for the nonrandom distribution of sex-biased genes, where in a male heterogametic system, male recessive beneficial alleles might be enriched in the X. On the other hand, for dominant female beneficial alleles, sitting on the X increases the time that the female beneficial allele spends as X-linked, putatively resulting in a higher effectiveness of positive selection acting on female beneficial loci (males X population size is reduced by 1/3)[14]. Surprisingly, we did not find a consistent pattern of sex-biased enriched genes on the X, but for male-biased genes in *L. splendidula* heads (Fig. 5B). Instead, we find certain autosomes to be enriched for either female- or male-biased chromosomes (Fig. 5B–E). Different degrees of dominance levels on sex-biased genes might explain autosomal enrichment of this gene class.

**Selection on sex-biased expressed genes**
In various organisms, faster protein evolutionary rates have been identified in sex-bias expressed genes and in X-linked genes, an observation that has been hypothesized to be caused by strong selective pressures caused by sexual selection, a relaxation of purifying selection or by the different effective population size of the X relative to the autosomes[12,44,45]. We looked at the variation of evolutionary rates between sex- and unbiased genes in the autosomes and the X-chromosome, using a branch-site model that allows the identification of variable evolutionary rates at a focal branch[46]. We did

not identify higher evolutionary rates between unbiased and sex-biased genes, nor between genes of different expression classes sitting on the autosomes or the X-chromosome (Fig. 6A, Fig. S8). For *L. italica*, no orthologous genes were found in the category of female-biased in head tissue. Sex-biased genes were less likely to have a blast hit between *L. splendidula* and *L. italica* (binary logistic regression, *P*-value = 2e-16, coefficient = −2.25), hinting at a more rapid sequence divergence between species within sex-biased genes, which could bias our branch-site model analysis toward a higher chance of analyzing genes with high inter-species sequence conservation.

We moved forward to investigate genomic patterns of nucleotide diversity ($\pi$)[47] in different genomic regions of sex- and unbiased expressed genes. To do so, we did whole genome re-sequencing of 15 individuals from the same population where the genome and RNA-seq individuals were collected. Based on our population genetic analysis, we found a higher nucleotide diversity in sex-biased genes than in unbiased genes, a pattern that was consistent in intronic regions (Fig. 6B, C), and which was often observed in promoters and exons.

## Discussion
Biological diversity offers us a wide variety of systems to tackle fundamental questions in evolutionary biology. Here, we used two different firefly species with different degrees of sexual dimorphism to start unraveling the genetic basis of sexually dimorphic traits. We assembled chromosome-level genomes of *L. splendidula* and *L. italica*, which diverged ~ 140 mya[48], showing different biological traits and whose genomes show divergent genomic characteristics (Table 1).

We successfully identified the X-chromosome in both species. Both X-chromosomes showed high chromosome synteny with each other as well as to other elateroid beetles (Fig. 2)[27,49]. The observed high X-chromosome conservation is quite different to what is observed in other systems such as fish or frogs, where high rates of sex chromosome turnover is described[50,51]. In insects, the age of the X chromosome, could go further back to 450 MY, as X-chromosome syntenic blocks have been found across nine insect orders, including true bugs, dragonflies, and grasshoppers[28]. Contrary to the high synteny conservation observed in the X-chromosome, we uncovered several autosomal fusion and fission events across five elaterid beetle species with divergent times up to ~ 180 mya. A small chromosome total count, as is the case of the analyzed species, could have a higher probability of fusion or

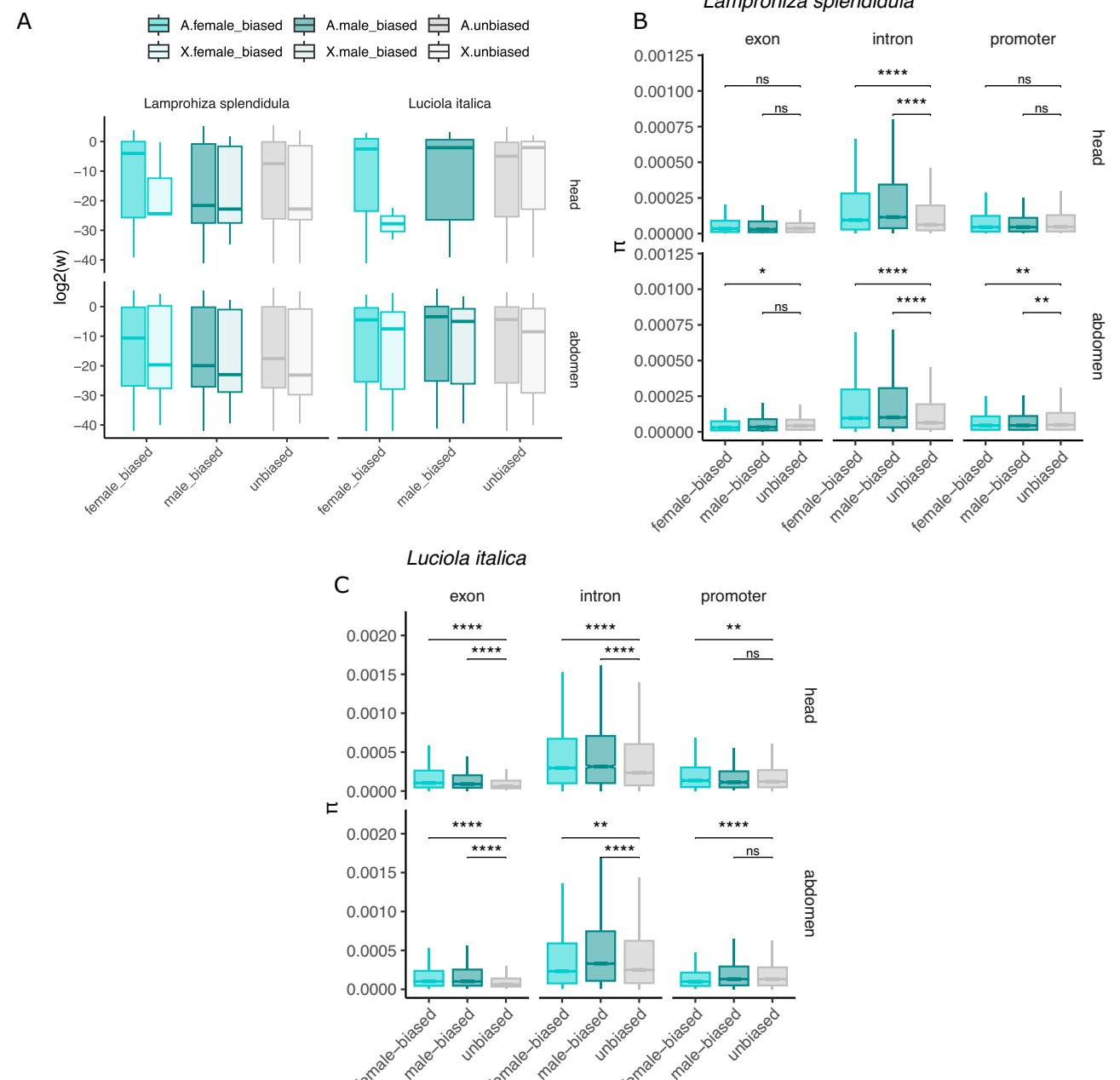

**Fig. 6 | Analysis of the selecting forces acting on sex-biased gene expression.**
**A** Branch-site model on sex-biased and unbiased expressed genes on the X and autosomes. Nucleotide diversity ($\pi$) distributions in (**B**) *L. splendidula* (**C**) *L. italica* in exons, introns and promoter regions. Significance between unbiased and male- or female-biased was assessed with a Wilcoxon rank-sum test. Significance levels: **** means *p*-value < 0.0001, *** means *p*-value < 0.001, ** means *p*-value < 0.01, ns means "not significant".

fissions events[52]. Homologous data from other firefly species, hint that the Y chromosome in our two fireflies is either absent or highly degraded[17,18]. The high conservation level of the X-chromosome within fireflies and the presence of DC in more divergent beetles such as in the flour beetle *Tribolium*, suggests a scenario where the mechanism for DC might be shared across beetles. Being beetles one of the most diverse taxa on earth[53], and acknowledging that only a handful of beetles have been explored for their DC status, we cannot rule out independent evolution of DC mechanisms in Coleoptera. Still, there is a possible scenario where a basal DC mechanism has evolved in Coleoptera and closely related groups[54], with putative divergent and species-specific adaptations of the DC machinery. We identified a significant overexpression of the X-chromosome in females and males of *L. italica*, hinting at the presence of an overexpression mechanism

to achieve equal expression between the autosomes and the X that still affects female's X expression. We do not observe an overexpression of the X-chromosome in *L. splendidula*, hinting that the female's X in this species is not affected by a male driven mechanism to achieve DC.

As *L. splendidula* and *L. italica* have different degrees of sexual dimorphism, we expected reflecting sex-biased gene expression. Indeed, in heads of the neotenic species, we observed a higher degree of sex-biased gene expression. *L. splendidula* female's heads are strongly reduced, including eyes and head capsule when compared to males. Surprisingly, in abdomens, we observed the reverse trend, where *L. italica* showed a higher degree of sex-biased expression. It is possible that spermatophore production in *L. italica*'s males accounts for the stronger sex-biased gene expression profile in this species. The presence of a spermatophore has been reported in other

*Luciola* species but not in *L. splendidula*[15]. There is very little information on physiological and anatomical differences across firefly species[2,55,56], information which needs to be further generated to have a more complete understanding of the observed gene expression divergence.

We identified many genes in heads and abdomens with strong sex-biased expression or even expressed in a sex-specific manner. Many of these genes are annotated as transposable elements (TE) (Supplementary data 1–4). TEs have been reported to be involved in the maintenance of phenotypic divergence within and between populations[43,57]. Many of the genes showing strong sex-biased expression levels have an annotation that suggests these to be involved in head development, flight capability and sex-tailored xenobiotic response (Fig. 4B). We uncovered adult female-specific expression of *luciferin-4* in *L. splendidula*. Why females express a sex-specific luciferase still needs to be elucidated, but hypotheses in the direction of mate choice, mating system or inter-species recognition can be explored.

Differences in selective pressures between the sexes can result in a non-random chromosomal distribution of sex-biased genes, these being often enriched in the sex chromosomes. We only found X-chromosome enrichment of male-biased genes in *L. splendidula*'s heads. Interestingly, we recovered a non-random distribution of sex-biased genes on various autosomes (Fig. 5B–E). Autosomal enrichment of sex-biased expressed genes reveals feminized or masculinized genomic regions. An enrichment of sex-biased genes in the autosomes could be the result of physical linkage or gene co-expression as part of the same functional network.

Sex-biased genes have often been identified to show faster evolutionary rates as opposed to unbiased expressed genes[58]. Despite these findings, we did not find faster evolutionary rates in sex-biased genes, as tested with a branch-site model. A lack of faster evolutionary rates as assessed by a branch-site model can be explained by (1) a complete resolution of fitness effects by sex-biased gene expression, which would fit with the lack of finding of enriched sex-biased genes on the X chromosome. Or, (2) a saturation of amino-acid changes due to the deep divergent times[59]. To rule out the later hypothesis, genetic information from closer related species could provide further insights. We detected a lower chance of finding a blast hit when a gene is sex-biased, which potentially indicates fast amino acid changes in sex-biased genes. In such a scenario, our branch-site model would be biased into analyzing more conserved genes between the two species.

Around 25% of sex-biased genes have lost or gained their sex-biased status between the two species (Fig. 5A). This result shows that most of the expressed genes showing sex-biased expression do not have this category in a divergent species, suggesting a high turnover rate of this gene class in fireflies. Genes that have incongruent sex-biased gene expression, might have acquired more recently a sex-biased expression status and therefore might be involved in sex- and species-specific phenotypes. We think that linage specific sex-biased gene expression has a higher chance of evolving through sexual selection. Despite the high turnover of sex-biased genes, we identified a percentage of genes showing conserved sex-biased expression between the two species. Conserved sex-biased gene expression has also been observed in other species, with variating numbers according to the developmental stage, tissue or species analyzed[8–10]. In fireflies, some of the genes showing female or male-biased expression in both species, might be involved in basal sexual dimorphic traits such as oviposition, sex-specific hormone receptors or conserved sex-specific neural development. These genes are of particular interest to yelp unveiling core networks that maintain sex-biased gene expression across species.

When we investigated nucleotide diversity patterns, measured as π, across different genomic regions, we uncovered significantly higher nucleotide diversity in sex-biased genes, a signal that was consistently observed in intronic regions. Higher π values are commonly linked to more variability being maintained at particular regions. Intronic regions have been found to play an important role in gene regulation controlling phenotypic variation, harboring *cis*-regulatory elements[60]. Forces such as balancing selection have been suggested to keep different alleles and thus inflate heterozygosity at particular sites[5]. The maintenance of two variants that

might be allele and sex-specific regulated might be a mechanism to achieve sex-biased gene expression. Divergent alleles between the sexes can also lead to sex-biased alternative splicing, where in various insects and vertebrates, alternative spliced transcripts are involved in sex-specific phenotypes[61].

## Conclusions

Fireflies are a diverse group of bioluminescent beetles presenting different degrees of sexual dimorphism. Novel organisms to investigate the evolution of sexually dimorphic traits, their genetic architecture, their sex-biased expression patterns and the evolutionary forces shaping these traits, are an invaluable resource that will enrich our understanding of the evolution of dimorphic sexes. The generation of two novel firefly genomes coupled with RNA-seq data, provided us with several insights: (1) We uncovered high synteny levels of the X-chromosome between *L. splendidula*, *L. italica* and other related beetles. (2) We report full X-chromosome dosage compensation in two divergent firefly species. (3) We uncovered many genes in heads and abdomens with strong sex-biased and sex-specific expression. These genes are primary candidates for sexually divergent phenotypes although we acknowledge that surveying gene expression at different developmental stages will contribute to solve the puzzle of gene expression to phenotypic causality. (4) We identified a set of sex-biased expressed genes with congruent expression between the two species. These genes might form part of key conserved gene networks involved in the maintenance of sexually dimorphic traits across species. (5) At the phylogenetic scale (branch-site model) we could not identify differential evolutionary rates in sex-biased genes, but at the population genetic scale, we uncovered higher nucleotide diversity patterns in sex-biased genes, especially in intronic regions, variation that could be maintained by differential sexual selection. Much more research needs to be done to further dissect the evolution of sexual dimorphism in fireflies, but this work will help us to start elucidating how sexual dimorphism is maintained across species.

## Methods

### Sample collection, nucleic acid extraction and sequencing

*L. splendidula* specimens were sampled in Munich (48.102443, 11.478317), Germany, and *L. italica* specimens were sampled in Lausanne (46.519540, 6.587329), Switzerland, using an insect net for winged individuals and by hand for neotenic females. After collection, the head and abdomen tissue was dissected and stored in DNA/RNA Shield at 4 °C for a week, and then transferred to −80 °C. The *Quick*-RNA Tissue/Insect Microprep kit (Zymo Research) was used to extract heads and bodies separately and concentration and quality were assessed with an Agilent Bioanalyzer. Illumina libraries and sequencing were done by Novogene, generating 150 bp paired-end sequences with 20x coverage for each sample.

DNA extractions were performed for a separate male individual collected in the same locations. For HWM DNA, the MagAttract HMW DNA kit (Qiagen) was used following the manufacturer's guidelines. DNA fragment sizes and integrity were checked with a 1% agarose gel and a Femto Pulse system (Agilent). Long DNA fragments were sequenced from a single male individual using Nanopore PromethION in a flongle cell and run for three days by SciLifeLab in Sweden. Illumina 150 bp paired-end reads were generated from the same individual and sequenced to a coverage of ~60x. Fifteen individuals were preserved in 96% ethanol from each population and extracted using the Monarch Genomic DNA Purification kit (New England BioLabs). DNA quality and integrity were assessed with a Nanodrop and an Agilent 5400. Illumina 150 bp paired-end reads were generated for each sample, aiming at 15x coverage. Library prep and sequencing was outsourced to Novogene, China.

### De novo genome assembly, scaffolding and annotation

For both species, the steps described below were taken. Base-calling of Nanopore reads was done with Guppy (4.0.11). Genome assembly was performed with Flye v2.8.1[62] and haplotype separation was done with purge_dups v1.2.6[63]. Two rounds of polishing were done, one with Medaka v1.7.2 (https://github.com/nanoporetech/medaka) using long Nanopore

reads and one using short Illumina reads with Hapo-G v1.3.2[64]. Genome size and genome heterozygosity levels were estimated with GenomeScope[65]. Sequences that do not belong to the class Insecta were identified using Blobtools v1.1.1[66] and removed from the assembly. Genome statistics were calculated with Quast v5.0.2[67] and genome completeness was assessed with BUSCO v5.2.2[68] using the dataset Insecta.

Libraries for HiC genome scaffolding were prepared from flash frozen tissue using the OmniC kit and further sequenced on an Illumina Nova-Seq600, where 150 bp paired-end reads at 60x were produced. Scaffolding and clustering of contigs into potential chromosomal groups was performed with 3D-DNA v201008[69] and Juicer v1.6[70]. Later, contig orientations were validated while ambiguous fragments were removed by manual curation using Juicebox v1.11.08 (https://github.com/aidenlab/Juicebox).

A species-specific repeat library was constructed with RepeatModeler v1.0.11 and genome masking was performed with RepeatMasker v 4.1.2[71] using the generated custom library. Structural annotation of protein coding genes was approached by combining gene sets from the three pipelines (1) GALBA v1.0.0[72]. BRAKER1 v2.1.6[73], and BRAKER2 v1.2.6[74], with TSEBRA v1.0.3[75]. GALBA was executed with miniprot v0.6[76] using the protein sequences of *Abscondita terminalis* downloaded from the NCBI website on December 21st 2022 (20493 sequences). The BUSCO v5.2.2[68] scores of the resulting gene set with endopterygota_odb10 are C:92.5% [S:88.1%, D:4.4%], F:3.8%,M:3.7%,n:2124. In order to generate evidence for running BRAKER1, HISAT2 version 2.2.1[77] was used to map the RNA-seq data of the five biological replicates, from both sexes and body parts, to the genome. In BRAKER1, GeneMark-ET v4.69[78] were used to generate a training gene set for AUGUSTUS v3.3.2[79], using the spliced alignment information of the RNA-seq data in GFF format, and AUGUSTUS generated a gene set with BUSCO scores C:93.5% [S:79.3%, D:14.2%], F:3.6%, M:2.9%,n:2124. For running BRAKER2, we used the Arthopoda partition of OrthoDB version 10.1[80] as evidence. Here, GeneMark-EP v4.69 was used to generate a training set for AUGUSTUS, and the final gene set was generated by AUGUSTUS. Here BUSCO scores were C:88.6% [S:84.3%, D:4.3%], F:5.5%, M:5.9%, n:2124. None of the gene set had a BUSCO completeness that satisfied our expectations. Therefore, we combined these gene sets with TSEBRA, enforcing all transcripts from the GALBA gene set. Proteins were functionally annotated with InterProScan 5.60–92.0[81]. We removed all single-exon genes without functional annotation by InterProScan from the final TSEBRA gene set, leading to BUSCO scores of C:95.5% [S:60.8%, D:34.7%], F:2.9%, M:1.6%, n:2124. In parallel, we performed a genome-guided transcriptome assembly using Trinity v2.8.5[82]. We noticed that transcript duplication was high (37.5%) after running BUSCO v5.2.2. We reduced this duplication to 4% by building "super transcripts" and collapsing unique and common sequence regions among splicing isoforms into a single linear sequence. This was done by running "Trinity_gene_splice_modeler.py" on the trinity-generated transcriptome FASTA file[83]. To recover the coordinates, the newly generated FASTA file was then mapped to the genome using minimap2 v2.14[84]. The output BAM file was then converted to GTF format with AGAT v0.8.1[85]. Afterward we merged the TSEBRA generated GTF file with the de-novo annotation GTF using "agat_sp_complement_annotations.pl" and removed all duplicated entries.

### *L. italica* phylogenetic placement

A high species coverage[16] subset of anchored hybrid enrichment (AHE) DNA fragments[27] was used to place *L. italica* in a phylogenetic context. The selected AHE set was mapped against the *L. italica* genome using minimap2 v2.14[37]. AHE hits in *L. italica* were extracted and aligned with its corresponding fragment using MAFFT v7.453[86]. Alignments were curated with TrimAl v 1.4.1[87]. Phylogenetic inference was done using RaxML v 8.2.11 implementing a GTRGAMMA model and performing 100 bootstraps[88].

### Population whole re-sequencing processing

Illumina short read sequences were trimmed with TrimmGalore! V0.6.6[89] and FastQC v0.11.9 were used to filter out bases with a phred score < 20 and reads shorter than 20 bp[90]. Reads were mapped to the genome with BWA

v0.7.17[91]. Mapped files in BAM format were curated by removing PCR duplicates with Picard v2.20.8, and low-quality reads (Q20) were discarded using SAMtools v1.10[92].

GATK v4.1.9[93] was used to call SNPs and indels by local reassembly of haplotypes with HaplotypeCaller. The joint genotyping of all sequenced samples was done with GenotypeGVCFs. VCFs statistics were drawn with bcftools stats and gatk VariantsToTable. Quality score thresholds were applied for minimum and maximum read depth [20,1568], fisher strand [FS = 10], strand bias [SOR = 3], root mean square mapping quality [MQ = 40] and nucleotide quality by depth [DP = 2]. Only variants with a QUAL > 30 were kept, as well as only SNPs (indels were removed) and biallelic sites. A SNP missingness of 0.25 was allowed across all samples. Sites in the VCF file that overlapped with repetitive elements were excluded from the analysis. An additional set of VCF files that included monomorphic sites was generated, where the GATK tag –select-type-to-include NO_VARIA-TION was used.

### Identification of the X chromosome

To identify the putative contigs belonging to the X chromosomes, we compared male to female (m:f) coverage ratio across contigs. *L. noctiluca* males are expected to be the heterogametic sex, thus we expect an m:f coverage ratio on the X to lie near 0.5. We used Illumina reads from 2 samples of each sex. Read quality control was done with FastQC v0.11.9 and adaptor and tail trimming was performed with Cutadapt v3.4 using a threshold of Phred <20[94]. The curated reads were mapped to the hard masked genome (Repeatmasker, v4.1.2) with BWA v0.7.17. Duplicate reads were removed from the bam files with Picard v2.20.8 (https://broadinstitute.github.io/picard/). Coverage was calculated with Deeptools v3.5.0 for 10Kb windows across the genome and normalized using RPKM (Reads Per Kilobase Million). Each 10 kb window coverage level was normalized by dividing it by the mean coverage value of the 5 five largest autosomal contigs. These five contigs were manually selected by choosing the five largest contigs with a male to female coverage ratio of 1 ± 0.1. Contigs smaller than 30 kb were filtered out leaving only contigs with at least 3 data points (i.e. three 10 kb windows). We then performed a nonparametric Wilcoxon rank sum test to test for significant differences in contig coverage values between sexes and applied a Bonferroni multiple test correction. The male to female coverage ratios were calculated only from contigs with significant differences in coverage between sexes. Contigs with an m:f ratio $0.4 \leq x \geq 0.6$ were considered to belong to the X chromosome.

### Genome synteny

For synteny analysis we additionally used the following three genomes *Photinus pyralis* [https://www.ncbi.nlm.nih.gov/datasets/genome/GCF_008802855.1/], *Cantharis rustica* [https://www.ncbi.nlm.nih.gov/datasets/genome/GCA_911387805.1/] and *Melanotus villosus* [https://www.ncbi.nlm.nih.gov/datasets/genome/GCA_963082815.1/]. We used satsuma2 [https://github.com/bioinfologics/satsuma2] to make pairwise whole genome alignments and visualization of genome synteny was done with NGenomeSyn[95].

### Gene expression analysis

RNA-seq reads were adaptor trimmed and quality filtered (–quality 20, –length 20) using Trimgalore v0.6.6[89]. Reads were mapped to the respective genome using Hisat2 v2.1.0[77] and FeatureCounts v.2.0.1[96] was used to produce a raw count matrix. As input GTF we used the Breaker-Augustus/Trinity merged annotation file generated as described above. EdgeR v3.34.1[97] was used to identify sex-biased expressed genes implementing a one factor analysis for each tissue type. Additionally, CPMs were calculated in EdgeR and used to calculate the f:m and A:X gene expression ratios. Orthologs between species was done through a reciprocal BLAST approach[98] using the generated genomes and the genome of *Photinus pyralis* (Ppyr1)[99].

We used TransDecoder v5.5.0[83] and Trinotate v.3.2.2[100] to obtain the likely proteins and annotate them in terms of known proteins and protein

domains. For this, the longest ORFs were extracted and queried against the Swiss-Prot database[101]. The functional domains were identified by mapping them to the PFAM domain database[102] using HMMER v3.3.2 (http://hmmer.org/). GO assignments were extracted from the Swiss-Prot/PFAM databases. Furthermore, unassigned ORFs were blasted against the NCBI non redundant protein database[103]. In all instances only the best hit was considered.

## Assessment of evolutionary rates

We performed a branch-site analysis with Godon (https://github.com/idavydov/godon-tutorial) to identify variating rates of evolutionary change across three linages: *L. splendidula*, *L. italica*, and *P. pyralis*, keeping only proteins derived from expressed transcripts. We used Prank (http://wasabiapp.org/software/prank/#Methods) to align coding sequences in a codon-aware manner. Alignments were curated using Gblocks[104]. To identify differences in nucleotide diversity between sex-biased and unbiased genes we calculated pairwise nucleotide diversity and Tajima's $D$ with VCFtools v0.1.14[105], separately for exons, introns, and promoter regions in sliding windows of 10000 base pairs. Differences between expression categories were evaluated with a Wilcoxon test.

## Reporting summary

Further information on research design is available in the Nature Portfolio Reporting Summary linked to this article.

## Data availability

We submitted all the data to NCBI and SRA. Detailed information on the SRA submission can be found in Supplementary Data 5. Genomes and annotation files can be found under the bioprojects PRJNA1083448 and PRJNA1090789.

## Code availability

All code and software sources used in our study are listed under the "Methods" section with corresponding citations of references.

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

## Acknowledgements

We would like to thank Sebastian Höhna, for his help during sample collection and support throughout the project, Gabriele Kumpfmüller, for her magical hands in the lab, Shanaka Thisara for imaging the *L. splendidula* specimens, Christian Tellgren-Roth and Remi Olsen from NGI Sweden for their bioinformatics advice, and Jochen Wolf for his ongoing support at the Evolutionary Biology Section. Funding for this study came from the Swiss agencies SVSN (Société Vaudoise des Sciences Naturelles) and SAV (Société Académique Vaudoise) to P.D. and from the Deutsche Forschungsgemeinschft (D.F.G.), SPP-2349 (CA 2207/4-1) to A.C.

## Author contributions

A.C. conceived the project. A.C., D.G., L.R., T.H., K.H. and P.D. performed data analysis. AC led the writing of the manuscript with input from the coauthors.

## Funding

## Competing interests

The authors declare no competing interests.
