## [Peer Review File · Communications Biology]

Reviewers' comments:

Reviewer #1 (Remarks to the Author):

In the paper by Catalan et al., the authors sequenced two firefly species and studied their sex-biased expression, dosage compensation, as well as a few other interesting topics. Similar to other beetles, it appears that these two firefly species exhibit full dosage compensation. Unlike some other insect species, they do not show patterns of X chromosome enrichment in sex-biased genes nor strong selective pressures on these genes. The paper is overall well-written, although there are some errors, especially in labeling figures. My primary comment is that the genome assemblies are major novelties in the paper; therefore, the genomes should be submitted to and annotated by NCBI or Ensembl before publication. NCBI usually offers a quick turnaround time.

Below are a few special comments:

1. The genomes and RNA-seq data should be submitted to NCBI. In the next version of the manuscript, please include the genome accession numbers. Additionally, it would be beneficial to use the NCBI annotated genome for some of the analysis to maintain consistency in genome versions. I understand the analyses are done and would be hard to reanalyze, but having a public genome is essential.
2. The authors described the assemblies as chromosome-level. How many chromosomes are assembled in one contig? If most chromosomes are not assembled in one contig, then they should not be referred to as chromosome-level. In some figures, the chromosomes seem to be completely assembled; in other cases, it looks like the genomes are still quite fragmented.
3. "The lower heterozygosity found in *L. splendidula* compared to *L. italica*" - How many individuals were sampled or sequenced in these species? Could the different sampling numbers or sample sources of the species influence the heterozygosity estimation?
4. Do the X chromosomes exhibit lower levels of heterozygosity in the two species? Although not absolute, in most species, the X chromosome typically has a lower amount of heterozygosity.
5. The authors mentioned, "Additionally, we verified the identity of the X chromosome by using homology to the soldier beetle *Cantharis rustica* (Figure S3)." However, Figure S3 only shows normalized coverages from sequencing. The X-axis in Figure S3 is also unclear.
6. Following the previous point, "Sex-specific genes were randomly distributed across the chromosomes, with the exception of *L. italica*'s heads, where sex-specific expressed genes were significantly enriched on the X chromosome (Fisher Exact Test: p-value 6.095e-08)." Later, it states, "Surprisingly, we did not find a consistent pattern of sex-biased enriched genes on the X, except for male-biased genes in *L. splendidula* heads (Figure xx)." The lack of a notable pattern in sex-specific expression on the X chromosome in the species is somewhat unexpected compared to other insect species, which the first author has extensive expertise. The authors tried to explain these patterns. Maybe I missed the information: what are the patterns of X expression and X-linked gene evolution in other beetles' species?
7. What does "congruent expression" mean? Do you mean a similar expression? How is this defined when comparing two highly divergent species?
8. There are errors in figure labeling throughout the document. Please check these carefully during revision.

Reviewer #2 (Remarks to the Author):

The authors build two high quality genomes for two species of fireflies, and use this to examine sex-biased gene expression as well as X chromosome conservation and dosage compensation. I have no real quibbles with the analysis other than the fact that I am unsure what exactly we learned from the study.

Sex-biased genes have been studied in a wide variety of animals, and comparative studies across gradients of sexual dimorphism have also been done. The authors state that the firefly species studied here have the potential to provide novel knowledge about the evolution of sexual dimorphism in this context, but they don't really reveal anything novel, as the results are, as far as I can tell, broadly concordant with all this previous work. I would encourage the authors to think carefully about what questions the fireflies allow us to ask that have not been addressed previously. This could be a result of their unique biology or mating system.

More broadly, I am somewhat concerned that the narrative of the intro, which presents the comparison of sex-biased genes in two species with large differences in sexual dimorphism, is somewhat at odds with the results, which really doesn't do this. Rather, the authors report that sex-biased expression is broadly similar between the species, aside from male based genes in the head. How this relates to sexual dimorphism is a bit unclear.

Otherwise, most of the results are largely confirmatory of previous work. Some of this is specific to Coleoptera, where a conserved X chromosome and X chromosome dosage compensation have been previously shown (<https://www.biorxiv.org/content/10.1101/2023.01.18.524646v1>). Other results have been shown in other animals, such as the turnover in sex-bias across related species. The authors are clear in all these things, and they cite and discussion the relevant literature.

Reviewer #3 (Remarks to the Author):

This paper describes the assembly and analysis of the genomes of two distantly related firefly species. To my knowledge, these are the first firefly genomes. Fireflies are a very interesting model for the study of the origin of new organs, sexual dimorphism and sexual selection, neoteny, and life history evolution. The genome assemblies appear to be of good quality with respect to completeness and contiguity. The authors have also generated a substantial amount of RNA sequence data that helps with genome annotation and serves as a valuable resource in its own right. Overall, the resources and analyses described in this paper should provide the background for future studies of firefly evolution, physiology, and behavior.

Compared to previously studied organisms, this paper does not report any novel or surprising conclusions. Rather, it replicates the patterns broadly observed in other models. The authors show convincing evidence for X-chromosome dosage compensation via $\sim 2x$ upregulation of the X in males. Dosage compensation occurs in most studied organisms with highly differentiated sex chromosomes, and X upregulation in males is one of the most common mechanisms. The study of sex-biased gene expression also produced predictable results: many genes show a weak sex bias while only a few are fully sex-specific. This analysis has the weakness shared by most previous studies in other organisms, in that gene expression was examined in complex mixtures of tissues (whole head and whole abdomens). Given the differences in organ composition between males and females (which are especially pronounced in the species with neotenic females), many if not most sex-biased gene expression is likely to reflect different abundance of most cell types in males vs females – but in truth, we don't know from this analysis how much "sex-biased gene expression" is due to sex-specific allometry, and how much to sex-specific gene regulation in each cell type. The differences between the

two species in which genes are sex-biased, and which are unbiased, are also not unexpected, as substantial turnover of sex-biased genes has been observed on much shorter time scales, e.g. between closely related species of *Drosophila* and *Caenorhabditis*. Overall, the novelty of this paper lies in providing a new resource and a springboard for future studies, rather than in its scientific conclusions.

A couple of questions and requests for clarification. First, has the reported number of chromosomes been confirmed by cytological karyotyping, or is it based solely on the genome assembly? The two assemblies differ in their contiguity. Chromosome 4 of *L. splendidula* looks like a clean rearrangement, but chromosomes 1 and (especially) 8 look suspicious. Can you rule out assembly artifacts? Second, please report the number of fragmented and duplicated BUSCO genes in Table 1, that helps with assessing assembly quality. And a minor thing – on line 270, figure number is missing.

RESPONSE

We appreciate the positive comments and openness of the reviewing process.

Referee expertise:

Referee #1: Insect evolutionary genomics

Referee #2: Evolutionary genomics and sex evolution

Referee #3: Insect evolutionary genomics, sex evolution

Reviewers' comments:

Reviewer #1 (Remarks to the Author):

In the paper by Catalan et al., the authors sequenced two firefly species and studied their sex-biased expression, dosage compensation, as well as a few other interesting topics. Similar to other beetles, it appears that these two firefly species exhibit full dosage compensation. Unlike some other insect species, they do not show patterns of X chromosome enrichment in sex-biased genes nor strong selective pressures on these genes. The paper is overall well-written, although there are some errors, especially in labeling figures. My primary comment is that the genome assemblies are major novelties in the paper; therefore, the genomes should be submitted to and annotated by NCBI or Ensembl before publication. NCBI usually offers a quick turnaround time.

Below are a few special comments:

1. The genomes and RNA-seq data should be submitted to NCBI. In the next version of the manuscript, please include the genome accession numbers. Additionally, it would be beneficial to use the NCBI annotated genome for some of the analysis to maintain consistency in genome versions. I understand the analyses are done and would be hard to reanalyze, but having a public genome is essential.

We have submitted the genomes and their corresponding annotation files, under the bioprojects PRJNA1083448 and PRJNA1090789. Detailed information on the SRA submission can be found in **Table S5**.

2. The authors described the assemblies as chromosome-level. How many chromosomes are assembled in one contig? If most chromosomes are not assembled in one contig, then they should not be referred to as chromosome-level. In some figures, the chromosomes seem to be completely assembled; in other cases, it looks like the genomes are still quite fragmented. We thank the reviewer for the opportunity to clarify the terminology. We call our assembly “chromosome level assembly” following the current terminology used for genome assemblies [6–8]. To our understanding, the successful placement of contigs into scaffolds and the identification of such scaffolds into chromosomes corresponds to a chromosome level assembly. Most of the literature available use technologies such as Hi-C to identify chromosome linkage of the contigs of interest, which is what we did [6–9] **Figure S1**. For *L. splendidula* and *L. italica* we recover a median length of ~ 65Mb for the identified chromosomes. These sizes correspond to almost fully completed chromosomes. We do still have several contigs in each of the genomes which could not be placed into a bigger

scaffold/chromosome. We hope that in future versions of the presented genomes this can be improved.

3. "The lower heterozygosity found in *L. splendidula* compared to *L. italica*" - How many individuals were sampled or sequenced in these species? Could the different sampling numbers or sample sources of the species influence the heterozygosity estimation? For the estimation of nucleotide diversity (heterozygosity π) we used whole genome re-sequencing Illumina data, 150 paired-end, ~15x coverage, from 23 individuals of the *L. italica* population and 20 individuals for *L. splendidula*. Thus, the number of individuals sampled between population was similar and evaluating diploid specimens gave us good confidence for our estimations for π . Additionally, we estimated heterozygosity levels using a k-mer approach. This analysis was done with GenomeScope (Figure R1).

Figure R1. Coverage histograms of *Luciola italica* (left) and *Lamprohiza splendidula* (right).

4. Do the X chromosomes exhibit lower levels of heterozygosity in the two species? Although not absolute, in most species, the X chromosome typically has a lower amount of heterozygosity.

Figure R2. Pairwise nucleotide diversity as calculated by π in the autosomes and X-chromosome.

For *L. splendidula* we do observe a lower pairwise nucleotide diversity on the X when compared to the autosomes. As the reviewer remarks, this is what is usually reported in the literature [10,11]. A lower nucleotide diversity on the X vs. the autosomes is expected due to a lower X population size, a higher chance of X-linked selective sweeps or sex-biased demographic events [11–13]. Contrary to expectations, the X-chromosome in *L. italica* shows a higher nucleotide diversity than the autosomes (**Figure R2**). Relaxed purifying selection in *L. luita* could explain this pattern. The X-chromosome of *L. italica* is 60% larger than that of *L. splendidula* but have similar gene content. It might be possible that a higher repeat content and/or intergenic regions than coding region are having an effect on inflating nucleotide diversity [14]. On the other hand, differences in selective pressures (purifying vs. positive selection) can affect nucleotide diversity patterns, between species and populations [15].

5. The authors mentioned, "Additionally, we verified the identity of the X chromosome by using homology to the soldier beetle *Cantharis rustica* (Figure S3)." However, Figure S3 only shows normalized coverages from sequencing. The X-axis in Figure S3 is also unclear. We are terribly sorry for this editing mistake. We have revised the manuscript and corrected it. We now refer to the figures shown in **Figure S4-S6** where chromosome homology across samples is addressed. In **Figures S4-S6** we can nicely see chromosome homology across *L. splendidula*, *L. italica* and *C. rustica*.

6. Following the previous point, "Sex-specific genes were randomly distributed across the chromosomes, with the exception of *L. italica*'s heads, where sex-specific expressed genes were significantly enriched on the X chromosome (Fisher Exact Test: p-value 6.095e-08)." Later, it states, "Surprisingly, we did not find a consistent pattern of sex-biased enriched genes on the X, except for male-biased genes in *L. splendidula* heads (Figure xx)." The lack of a notable pattern in sex-specific expression on the X chromosome in the species is somewhat unexpected compared to other insect species, which the first author has extensive expertise. The authors tried to explain these patterns. Maybe I missed the information: what are the patterns of X expression and X-linked gene evolution in other beetles' species? Thanks for bringing our attention to this. We realize how those two sentences might be confusing. We have now edited this text to make it clearer (lines 183-189). Briefly, we performed two analyses, (1) we checked on the chromosomal distribution of sex-specific genes, pooling female and male-specific genes together, mainly to have a higher number of observations for statistical testing. (2) We investigated the presence of chromosomal enrichment of sex-biased genes (regardless of whether these are sex-specific or not). Sex-specific genes are genes that are only expressed in one sex, as opposed to sex-biased genes, where these genes are expressed in both sexes but at significantly different levels. Genes with a sex-specific expression might have the potential of being very detrimental for the other sex, therefore a sex-specific expression could resolve this conflict. In the case of male-specific (this also applies to sex-biased genes) expressed genes, these might be enriched on the X chromosome if recessive. In the case of female-specific expressed genes, if dominant, these genes could be fixed in the X-chromosome. We found only in *L. italica*'s heads, an excess of sex-specific expressed genes on the X. From the literature that we have revised and from the data that the first author has analyzed in the past (*Drosophila*, *Heliconius* and European crow), sex-specific genes are rare to find, but also depends on the species [16]. Additionally, finding a distribution bias also depends on the organ that is being tested as well as the species. Depletion of male-biased genes on the X have been found in gonads, of *Drosophila*, *Timema* or *Tribolium* [17–19], whereas in *Anopheles* a paucity of male-biased genes was only observed in some *Anopheles* species [16]. In our case, we did not dissect gonads but analyzed whole abdomens. It is possible, that when analyzing single tissues, the chromosomal distribution pattern of sex-biased genes might show some shifts. On the other hand, a depletion of male-biased genes on

the X chromosome has also been linked with a germline specific X inactivation which can be detrimental to males [20]. A male-specific germline X inactivation has only been identified in *D. melanogaster*, mice and humans, and it is highly probable that is not a conserved phenomenon across animals [21].

7. What does "congruent expression" mean? Do you mean a similar expression? How is this defined when comparing two highly divergent species?

We sought to explore how conserved sex-biased gene expression is between two divergent firefly species. For this, we identified 1:1 orthologous genes between the two species and assessed their sex-biased status. The orthologous genes showing female or male-biased expression in both species, were classified as having congruent expression. The term congruent was also used when orthologous genes were unbiased in both species. On the other hand, orthologous genes showing incongruent sex-biased expression, show unbiased expressed in one species but sex-biased in the other. Genes female-biased expressed in one species but male-biased expressed in the other (or vice versa), were also classified as genes with incongruent expression.

8. There are errors in figure labeling throughout the document. Please check these carefully during revision.

We are terribly sorry about this. We have revised the manuscript and made the required edits.

Reviewer #2 (Remarks to the Author):

The authors build two high quality genomes for two species of fireflies, and use this to examine sex-biased gene expression as well as X chromosome conservation and dosage compensation. I have no real quibbles with the analysis other than the fact that I am unsure what exactly we learned from the study.

Sex-biased genes have been studied in a wide variety of animals, and comparative studies across gradients of sexual dimorphism have also been done. The authors state that the firefly species studied here have the potential to provide novel knowledge about the evolution of sexual dimorphism in this context, but they don't really reveal anything novel, as the results are, as far as I can tell, broadly concordant with all this previous work. I would encourage the authors to think carefully about what questions the fireflies allow us to ask that have not been addressed previously. This could be a result of their unique biology or mating system.

We thank the reviewer for bringing our attention to this and letting us clarify our scientific questions. For this study we chose two firefly species with two different degrees of sexual dimorphism (mild and extreme) to start understanding the gene expression regulation involved in sex-biased traits, such as head capsule size, eye size, wing size and overall abdomen size. We indeed, found many genes with strong sex-biased expression, which seem to be good candidates and putative key players in species-specific maintenance of sexually dimorphism. The identification of such genes is the first step into moving forward with functional work. Specially interesting are sex-specific expressed genes, since these most probably regulate processes exclusive of one sex. Much more work will need to be done to connect the genetic background + gene expression profiling to mating types, the presence of spermatophores or other type of organ or cell differences between the sexes and across species. Thus, this study provides gene expression profiling comparing two species with different types of sexual dimorphism, research that has not been done until now.

Beside the identification of sex-biased genes specific to each species, we were also able to identify a set of genes whose orthologous genes were sex-biased expressed in both. These genes are of particular interest since these could help unveiling core networks that maintain sex-biased gene expression across species.

Additionally, we revealed complete dosage in the single male X of both species, hinting that DC might be conserved in Lampyridae. Even though, complete dosage compensation has been reported for other coleopterans (*Tribolium spp*) we should not assume that this result can be extrapolated to other species, since beetles form one of the most diverse groups and present deep diverged times.

More broadly, I am somewhat concerned that the narrative of the intro, which presents the comparison of sex-biased genes in two species with large differences in sexual dimorphism, is somewhat at odds with the results, which really doesn't do this. Rather, the authors report that sex-biased expression is broadly similar between the species, aside from male based genes in the head. How this relates to sexual dimorphism is a bit unclear.

Thanks for pointing out the disconnect between the introduction, the results section and some parts of the discussion. We have revised these sections, and we hope now the manuscript connectiveness is improved. We found that most sex-biased genes with a 1:1 orthologue show incongruent sex-biased expression (**Figure 5A**). This result poses the hypothesis of high turnover of sex-biased expressed genes. On the other hand, we also find a percentage of orthologous genes that show the same direction of sex-biased gene expression. The percentage of conserved sex-biased expression across species varies depending on the phylogeny and tissue investigated [22–24]. We discuss about the possibility that these conserved genes might be involved in joint and conserved mechanisms maintaining sex-biased gene expression. We have cited and discussed this literature in the main text.

Otherwise, most of the results are largely confirmatory of previous work. Some of this is specific to Coleoptera, where a conserved X chromosome and X chromosome dosage compensation have been previously shown (<https://www.biorxiv.org/content/10.1101/2023.01.18.524646v1>). Other results have been shown in other animals, such as the turnover in sex-bias across related species. The authors are clear in all these things, and they cite and discussion the relevant literature.

Reviewer #3 (Remarks to the Author):

This paper describes the assembly and analysis of the genomes of two distantly related firefly species. To my knowledge, these are the first firefly genomes. Fireflies are a very interesting model for the study of the origin of new organs, sexual dimorphism and sexual selection, neoteny, and life history evolution. The genome assemblies appear to be of good quality with respect to completeness and contiguity. The authors have also generated a substantial amount of RNA sequence data that helps with genome annotation and serves as a valuable resource in its own right. Overall, the resources and analyses described in this paper should provide the background for future studies of firefly evolution, physiology, and behavior.

We thank the reviewer for this positive comment.

Compared to previously studied organisms, this paper does not report any novel or surprising conclusions. Rather, it replicates the patterns broadly observed in other models. The authors show convincing evidence for X-chromosome dosage compensation via ~2x upregulation of the X in males. Dosage compensation occurs in most studied organisms with highly differentiated sex chromosomes, and X upregulation in males is one of the most common mechanisms. The study of sex-biased gene expression also produced predictable results: many genes show a weak sex bias while only a few are fully sex-specific. This analysis has the weakness shared by most previous studies in other organisms, in that gene expression was

examined in complex mixtures of tissues (whole head and whole abdomens). Given the differences in organ composition between males and females (which are especially pronounced in the species with neotenic females), many if not most sex-biased gene expression is likely to reflect different abundance of most cell types in males vs females – but in truth, we don't know from this analysis how much “sex-biased gene expression” is due to sex-specific allometry, and how much to sex-specific gene regulation in each cell type. The differences between the two species in which genes are sex-biased, and which are unbiased, are also not unexpected, as substantial turnover of sex-biased genes has been observed on much shorter time scales, e.g. between closely related species of *Drosophila* and *Caenorhabditis*. Overall, the novelty of this paper lies in providing a new resource and a springboard for future studies, rather than in its scientific conclusions.

We certainly understand the reviewer's concern on what is being measured in the RNA-seq experiment, whether it is differences in gene expression or differences in tissue types or in cell abundances between the sexes. One of the main objectives of this study was to interrogate sex-biased gene expression in two divergent species and identify the identity of these genes and their expression difference. Even though, we do not have more specific information on the cell types where the identified sex-biased genes are expressed, we can already take a candidate gene approach to further investigate sexual dimorphic traits. In the near future we wish to generate more detailed information in order to pin point the specific source of sex-biased expressed genes.

A couple of questions and requests for clarification. First, has the reported number of chromosomes been confirmed by cytological karyotyping, or is it based solely on the genome assembly? The two assemblies differ in their contiguity. Chromosome 4 of *L. splendidula* looks like a clean rearrangement, but chromosomes 1 and (especially) 8 look suspicious. Can you rule out assembly artifacts? Second, please report the number of fragmented and duplicated BUSCO genes in Table 1, that helps with assessing assembly quality. And a minor thing – on line 270, figure number is missing.

The estimated number of chromosomes was done during the process of Hi-C scaffolding and we cannot rule out the presence of artifacts during the Hi-C scaffolding process or genome assembly. We expect that genome assembly artifacts and annotation mistakes can be improved and be further curated in the future. Until now we have not performed cytological karyotyping. For this, we would have to collect fresh specimens and preserve the tissue adequately for karyotyping analysis [25]. We hope to produce cytological karyotype information in the near future.

We have added the requested BUSCO statistics in **Table 1**.

We are terribly sorry for the editing mistakes. We have revised and corrected these.

References

1. Coenye T. 2021 Do results obtained with RNA-sequencing require independent verification? *Biofilm* **3**, 100043. (doi:10.1016/j.biofilm.2021.100043)
2. Müller L, Hutter S, Stamboliyska R, Saminadin-Peter SS, Stephan W, Parsch J. 2011 Population transcriptomics of *Drosophila melanogaster* females. *BMC Genomics* **12**, 81. (doi:10.1186/1471-2164-12-81)
3. Catalán A, Hutter S, Parsch J. 2012 Population and sex differences in *Drosophila melanogaster* brain gene expression. *BMC Genomics* **13**, 654. (doi:10.1186/1471-2164-13-654)
4. Pombo MA, Zheng Y, Fei Z, Martin GB, Rosli HG. 2017 Use of RNA-seq data to

- identify and validate RT-qPCR reference genes for studying the tomato-Pseudomonas pathosystem. *Sci. Rep.* **7**, 1–11. (doi:10.1038/srep44905)
5. Bustin S, Huggett J. 2017 qPCR primer design revisited. *Biomol. Detect. Quantif.* **14**, 19–28. (doi:10.1016/j.bdq.2017.11.001)
 6. Karimi K, Do DN, Wang J, Easley J, Borzouie S, Sargolzaei M, Plastow G, Wang Z, Miar Y. 2022 A chromosome-level genome assembly reveals genomic characteristics of the American mink (*Neogale vison*). *Commun. Biol.* **5**, 1–11. (doi:10.1038/s42003-022-04341-5)
 7. Hu QL, Ye ZX, Zhuo JC, Li JM, Zhang CX. 2023 A chromosome-level genome assembly of *Stenchaetothrips biformis* and comparative genomic analysis highlights distinct host adaptations among thrips. *Commun. Biol.* **6**. (doi:10.1038/s42003-023-05187-1)
 8. Zhang L *et al.* 2020 Chromosome-level genome assembly of the predator *Propylea japonica* to understand its tolerance to insecticides and high temperatures. *Mol. Ecol. Resour.* **20**, 292–307. (doi:10.1111/1755-0998.13100)
 9. Chen M *et al.* 2021 A chromosome-level assembly of the harlequin ladybird *Harmonia axyridis* as a genomic resource to study beetle and invasion biology. *Mol. Ecol. Resour.* **21**, 1318–1332. (doi:10.1111/1755-0998.13342)
 10. Vicoso B, Charlesworth B. 2006 Evolution on the X chromosome: unusual patterns and processes. *Nat. Rev. Genet.* **7**, 645–653. (doi:10.1038/nrg1914)
 11. Nam K *et al.* 2015 Extreme selective sweeps independently targeted the X chromosomes of the great apes. *Proc. Natl. Acad. Sci. U. S. A.* **112**, 6413–6418. (doi:10.1073/pnas.1419306112)
 12. Vicoso B, Charlesworth B. 2006 Evolution on the X chromosome: unusual patterns and processes. *Nat. Rev. Genet.* **7**, 645–653. (doi:10.1038/nrg1914)
 13. Stephan W, Li H. 2007 The recent demographic and adaptive history of *Drosophila melanogaster*. *Heredity (Edinb)*. **98**, 65–68. (doi:10.1038/sj.hdy.6800901)
 14. Chen ZH, Zhang M, Lv FH, Ren X, Li WR, Liu MJ, Nam K, Bruford MW, Li MH. 2018 Contrasting patterns of genomic diversity reveal accelerated genetic drift but reduced directional selection on X-chromosome in wild and domestic sheep species. *Genome Biol. Evol.* **10**, 1282–1297. (doi:10.1093/gbe/evy085)
 15. Casillas SS, Barbadilla A. 2017 Molecular Population Genetics. *Genetics* **205**, 1003–1035. (doi:10.1534/genetics.116.196493)
 16. Papa F, Windbichler N, Waterhouse RM, Cagnetti A, Amato D, Persampieri T, Lawniczak MKN, Nolan T, Papathanos PA. 2016 Rapid evolution of female-biased genes among four species of *Anopheles malaria* mosquitoes Running title : Evolution of sex-biased genes in *Anopheles*. *Genome Res.* , 1–50. (doi:10.1101/gr.217216.116)
 17. Zhang Y, Sturgill D, Parisi M, Kumar S, Oliver B. 2007 Constraint and turnover in sex-biased gene expression in the genus *Drosophila*. *Nature* **450**, 233–7. (doi:10.1038/nature06323)
 18. Parker DJ, Jaron KS, Dumas Z, Robinson-Rechavi M, Schwander T. 2022 X chromosomes show relaxed selection and complete somatic dosage compensation across *Timema* stick insect species. *J. Evol. Biol.* **35**, 1734–1750. (doi:10.1111/jeb.14075)
 19. Bracewell R, Tran A, Chatla K, Bachtrog D. In press. Sex chromosome evolution in beetles. (doi:10.1101/2023.01.18.524646)
 20. Kemkemer C, Catalán A, Parsch J. 2014 ‘Escaping’ the X chromosome leads to increased gene expression in the male germline of *Drosophila melanogaster*. *Heredity (Edinb)*. **112**, 149–55. (doi:10.1038/hdy.2013.86)
 21. Meiklejohn CD, Landeen EL, Cook JM, Kingan SB, Presgraves DC. 2011 Sex Chromosome-Specific Regulation in the *Drosophila* Male Germline But Little

- Evidence for Chromosomal Dosage Compensation or Meiotic Inactivation. **9**.
(doi:10.1371/journal.pbio.1001126)
22. Naqvi S, Godfrey AK, Hughes JF, Goodheart ML, Mitchell RN, Page DC. 2019 Conservation, acquisition, and functional impact of sex-biased gene expression in mammals. *Science (80-.)*. **365**. (doi:10.1126/science.aaw7317)
 23. Khodursky S, Svetec N, Durki SM, Zhao L. 2020 The evolution of sex-biased gene expression in the Drosophila brain. *Genome Res*. **30**, 874–884.
(doi:10.1101/gr.259069.119)
 24. Rodríguez-Montes L, Ovchinnikova S, Yuan X, Studer T, Sarropoulos I, Anders S, Kaessmann H, Cardoso-Moreira M. 2023 Sex-biased gene expression across mammalian organ development and evolution. *Science (80-.)*. **382**, eadf1046.
(doi:10.1126/science.adf1046)
 25. Dias CM, Schneider MC, Rosa SP, Costa C, Cella DM. 2007 The first cytogenetic report of fireflies (Coleoptera, Lampyridae) from Brazilian fauna. *Acta Zool*. **88**, 309–316. (doi:10.1111/j.1463-6395.2007.00283.x)

REVIEWERS' COMMENTS:

Reviewer #1 (Remarks to the Author):

The authors addressed all my comments.

Reviewer #3 (Remarks to the Author):

I appreciate the changes and clarifications the authors have made in their revised manuscript. I will leave it to the other reviewers to judge whether the details provided in the revised manuscript and in the response letter address their concerns. My own evaluation of this manuscript remains unchanged. On the one hand, the genome assemblies and the other data generated by the authors establish a valuable resource for future evolutionary and developmental-genetic studies in this interesting model, and the descriptive analyses reported in the paper will help jump-start these future studies. At the same time, the paper itself does not report any novel biological findings or provide any new insights into the unusual biology of these organisms. For the most part, it replicates the well-described general patterns seen in many other models without deepening our understanding of sexual dimorphism, dosage compensation, neoteny, light organs, etc. As before, I think the main value of this paper lies in the resources it describes, not in its scientific findings.

RESPONSE

We are happy that we could respond satisfactorily most of the comments suggested by the editor and reviewers.

REVIEWERS' COMMENTS:

Reviewer #1 (Remarks to the Author):

The authors addressed all my comments.

Reviewer #3 (Remarks to the Author):

I appreciate the changes and clarifications the authors have made in their revised manuscript. I will leave it to the other reviewers to judge whether the details provided in the revised manuscript and in the response letter address their concerns. My own evaluation of this manuscript remains unchanged. On the one hand, the genome assemblies and the other data generated by the authors establish a valuable resource for future evolutionary and developmental-genetic studies in this interesting model, and the descriptive analyses reported in the paper will help jump-start these future studies. At the same time, the paper itself does **not report any novel biological findings or provide any new insights into the unusual biology of these organisms**. For the most part, it replicates the well-described general patterns seen in many other models without deepening our understanding of sexual dimorphism, dosage compensation, neoteny, light organs, etc. As before, I think the main value of this paper lies in the resources it describes, not in its scientific findings.

We appreciate the point of view of the reviewer. We think that we provide novel and interesting insights into sex-biased gene expression evolution in a system where this type of research has not been done before. The identification of sex-biased genes constitutes the first step into moving forward with functional work. Thus, this study provides gene expression profiling comparing two species with different types of sexual dimorphism and deep divergent times, research that has not been done until now. Beside the identification of sex-biased genes specific to each species, we were also able to identify a set of genes whose orthologous genes were sex-biased expressed in both. These genes are of particular interest since these could help unveiling core networks that maintain sex-biased gene expression across species. Additionally, we revealed complete dosage in the single male X of both species, hinting that DC might be conserved in Lampyridae. Even though, complete dosage compensation has been reported for other coleopterans (*Tribolium spp*) we should not assume that this result can be extrapolated to other species, since beetles form one of the most diverse groups and present deep diverged times.